# Evaluation of a Configurable, Mobile and Modular Floor-Pen System for Group-Housing of Laboratory Rabbits

**DOI:** 10.3390/ani11040977

**Published:** 2021-04-01

**Authors:** Dana Matzek, Hanna-Mari Baldauf, Rico Schieweck, Bastian Popper

**Affiliations:** 1Biomedical Center, Core Facility Animal Models, Faculty of Medicine, Ludwig-Maximilians-Universität Munich, Großhaderner Straße 9, 82152 Planegg-Martinsried, Germany; Dana.Matzek@med.uni-muenchen.de; 2Max von Pettenkofer Institute & Gene Center, Virology, National Reference Center for Retroviruses, Faculty of Medicine, Ludwig-Maximilians-Universität Munich, 80336 Munich, Germany; Baldauf@mvp.lmu.de; 3Biomedical Center, Anatomy and Cell biology, Faculty of Medicine, Ludwig-Maximilians-Universität Munich, Großhaderner Straße 9, 82152 Planegg-Martinsried, Germany; Rico.Schieweck@med.uni-muenchen.de

**Keywords:** 3R strategy, animal welfare, New Zealand White

## Abstract

**Simple Summary:**

The group housing of animals supports species-specific behavior and avoids stress induction. Therefore, novel housing systems need to be established that provide sufficient space and meet the hygiene criteria for laboratory animals. Here, we describe the successful implementation of an improved housing system for single and group housing of laboratory rabbits under high hygiene environmental conditions. This system includes modular and mobile floor pens of different size and shape. Rabbits housed in this system showed species-specific behavior that depended on the circadian rhythm—a prerequisite for interpretable animal experiments. We propose that this housing system will be of great benefit for the well-being of laboratory rabbits used in biomedical science.

**Abstract:**

The major responsibility of researchers and laboratory animal facilities is to ensure animal well-being during the time of acclimatization, experiments, and recovery. In this context, animal housing conditions are of utmost importance. Here, we implemented a mobile and modular floor-pen housing system for laboratory rabbits that combines rabbits’ natural behavioral requirements and the high hygiene standards needed in biomedical science. Twelve female New Zealand White (NZW) rabbits were single- or group-housed for 12 months in mobile and modular floor-pens. Their general health status was evaluated at the end of the experimental setup. Further, we performed behavioral analysis of six additional NZW females group-housed for eight weeks in pens of two different sizes. We show that our improved housing concept supported species-specific behavioral patterns. Taken together, our housing system provides an optimal setup for rabbits in animal facilities that combines strict requirements for animal experiments with animal welfare.

## 1. Introduction

The concept of refinement, reduction, and replacement, also called the 3R strategy, is an intrinsic part of the EU Directive 2010/63 to protect all non-human vertebrates used for scientific purposes [1]. In this context, housing strategy is an important aspect that needs to be considered. The single housing of animals tends to lead to stereotypic and stress-related behavior [2,3,4]. Traditional cage housing of rabbits, even in pairs, is often accompanied by bone deformation, spinal and hip injuries, and behavioral abnormalities most likely caused by space limitations [5]. Moreover, space for species-specific behavior is a prerequisite to withdraw or to maintain inter-individual distances, which in turn allows hierarchy formation without intense fights and stress [4]. These aspects are particularly important for highly social species, such as the European rabbit (*Oryctolagus cuniculus*). Therefore, the adequate group housing of male and female rabbits has attracted extensive attention over the past decades [5,6,7,8].

It has been recommended that the housing of laboratory animals should ensure sufficient space to permit species-typical locomotion, explorative behavior, and social interactions. In this context, direct social interaction, or at least social non-contact enrichment, is required for animals’ well-being [9]. Thus, social interactions are a prerequisite even in conventional cages [10] and are known to impact the outcomes of research experiments [4,11]. In line with this notion are behavioral studies of adult New Zealand White (NZW) rabbits. Here, it has been shown that they prefer group housing over solitary living [2,12]. Consequently, the group housing of animals has to ensure the establishment of stable social orders, sufficient space for withdrawal, and an enriched environment allowing species-specific behavior to prevent stress, aggression, and injuries [2,3,4].

Importantly, group-housing strategies have to meet the criteria for the specific-pathogen-free (SPF) housing of laboratory animals. These standardized environmental housing conditions should support the animals’ natural, circadian-driven behavior. The circadian rhythm has an impact on a broad range of physiological, circadian-controlled parameters, especially behavior, that might impact the outcome of research experiments [13].

Thus, it is important to establish new group-housing systems that are applicable for laboratory animals. In our study, we aimed at implementing and evaluating a modular and mobile floor-pen housing system for rabbits under routine laboratory settings. Therefore, we tested NZW rabbits for their health status and species-specific behavior. We report that our housing system allows for species-specific behavior that depends on the circadian cycle. Moreover, our system is applicable for SPF conditions in animal facilities and provides the possibility to flexibly adjust size restrictions according to, for example, population size or official rules. Therefore, the modular floor pen system represents a refinement of animal housing and an alternative to conventional single or pair cage-housing systems.

## 2. Materials and Methods

### 2.1. Animals

Eighteen outbred New Zealand White (NZW) female rabbits (Crl:KBL), aged between 1 to 4 years and body weights between 1.8 to 5.5 kg, were purchased from Charles River laboratories (CR, France) and housed under barrier conditions in a specific-pathogen-free (SPF) animal facility.

### 2.2. Laboratory Environment

Room temperature and relative humidity were 18–20 °C and 45–55%, respectively. The light cycle was adjusted to 12 h light and 12 h dark periods. Room air was exchanged 15 times per hour and filtered via a high-efficiency particulate air (HEPA) filter system. All rabbits had free access to water (semi-desalinated water) and food (irradiated 4 mm pellet; Altromin 2120, Altromin Germany, Lage, Germany) as well as irradiated and autoclaved hay (Ssniff, Soest, Germany). Soiled bedding (fs14, Abedd, Wien, Austria) was removed each day and partially replaced when necessary. Pens were sanitized (rack washer for mouse individually-ventilated-cage systems) and decontaminated (fumigation by H_2_O_2_). Pens were completely exchanged once per week. Hygiene monitoring was performed every six months based on the recommendations of the Federation of European Laboratory Animal Science Associations (FELASA) working group [14]. Health status was evaluated every four weeks during routine experimental treatments. At the end of the study, a complete gross necropsy examination of all rabbits was performed after euthanasia by injection of a lethal dose of sodium-pentobarbital (Narcoren, Boehringer Ingelheim (100 mg/kg)). All experiments were in accordance with the local government authorities (Government of Upper Bavaria Az.55.2-1-54-2532.0-85-2016) as well as European (RL2010/63EU) and German animal welfare legislation. Furthermore, the experiments have been approved by the institutional ethical review committee (LMU Munich, Biomedical Center, Core facility animal models) and were conducted under the license Az.5.1-5682 (LMU/BMC/CAM).

### 2.3. Housing 

The dimensions of a single mobile-R-pen module are as follows (see Figure 1A,B): Length (L)—160 cm, Width (W)—80 cm, Height (H)—68 (front) to 108 cm (back). Maximum floor space of a single pen: 12,800 cm^2^ = 1.28 m^2^ (recommendations of the European animal welfare legislation (RL2010/63 EU) for rabbits ranging from 3 to 5 kg (minimum: 4200–5400 cm^2^)) [1]. The scaffold of the pen can be easily adjusted to the animals’ weight and laboratory requirements (Figure 1A,C). Disassembly of partition walls and enrichment items allows proper decontamination of all surfaces by H_2_O_2_ (Figure 1D,E).

#### 2.3.1. Single mobile-R-pen

Each module consists of five partition walls (two translucent side walls, height 68 cm—allows visual contact of animals), three black backplanes (108 cm) (see Figure 1D,E), and an elevated platform on the back wall (L65.5 cm × W25.0 cm × H29.0 cm). To provide a complex environment, tubes and tents made by the research team were added to each module and served as hiding places. Wood bricks (Plexx, The Netherlands) were added to each pen and replaced every month. Hay was provided as loose material on the floor or in nets to stimulate chewing behavior. Bedding material was piled in each pen with at least 2 cm height to stimulate burrowing behavior (Figure 2A–C). The solid floor also allows the use of additional bedding material, such as hemp mats, to enrich the floor structure (Figure 2C). All enrichment materials were either autoclaved or fumigated by hydrogen peroxide (H_2_O_2_) to ensure thorough decontamination. Notably, the SPF status of the colony was stable during the study, indicating adequate and appropriate decontamination measures for the floor-pen elements. Each module can be rotated by 90° or 180°. This assembly strategy provides different and complex environmental setups that stimulate the explorative behavior of rabbits. Furthermore, single pens can be easily connected to a larger area (Figure 2B,C). By adding the transparent partition walls, animals can be quickly separated without handling. Moreover, the mobility of the R-pen provides greater flexibility in room management. Each floor pen is accessible through a door at the front for easier handling of the rabbits (Figure 2A,C).

#### 2.3.2. Small mobile-R-pens

Three mobile-R-pen modules in a row were connected by removing the opaque subdividing walls. By doing so, we could create a maximum floor size of 3.84 m^2^.We provided all enrichment items as described for the standard pen (Figure 2B,C).

#### 2.3.3. Large mobile-R-pens

A total of six mobile-R-pen modules in a row were connected by removing the opaque subdividing walls in between each module to form a 7.68 m^2^ large enclosure. We provided all enrichment items as described for the standard pen (Figure 2B,C).

### 2.4. Procedure

#### 2.4.1. Health Study

Twelve adult female (2–4 years old) rabbits (weight: 2.8 to 5.5 kg), were randomly allocated in either three groups of at least two animals (2 × 2 + 3 animals housed in at least three pens in a row) or five individuals (housed in single pens next to each other) and monitored for one year. For all experiments, we compared a group of seven animals housed in three pens to a group of five animals in six pens. To determine the impact of housing on their behavior under routine lab work conditions, we chose rabbits that were used for antibody production and were handled once every four weeks for subcutaneous injections and/or bleedings over a longer period of time. In addition, we examined housing-related health problems, such as bone deformation or paw lesions, at the end of the experiments using gross necropsy in all twelve female NZW rabbits. Testing the group-housing effect on the efficacy of antibody production was not included in the study since this has been previously reported [3,5].

#### 2.4.2. Behavioral Study

We performed behavioral analyses for 8 weeks using six young adult (1–2 years old) rabbits (weight: 1.8–2.1 to 2.8–3.6 kg) that were not included in the health study. We decided to use six animals for the behavioral analyses because group sizes ranging from four to eight animals have been recommended for group housing in animal facilities [4]. To address the question of whether spatial restriction leads to behavioral alterations, a continuous scan sampling technique was used to analyze the animals’ behavior in a large floor-pen area (6-pen trial—max. floor size 7.68 m^2^) and a smaller area (3-pen trial—max. floor size 3.84 m^2^). The rabbit ethogram (Table 1) was adapted from previous publications on NZW rabbits’ behavior in cages and pens [6,15,16]. The behavioral analyses were conducted on a daily basis at the same time points (7:30 a.m., 9:30 a.m., 11:30 a.m., and 12:30 p.m.) by the same observer. Manual data collection (30 min observation per time point) was combined with video recording to allow retrospective analyses. The colony was familiarized to the 6-pen setup on delivery day. Animals were monitored for three days between 7:30 and 8:00 a.m. to evaluate the baseline behavioral patterns during colony formation. Rabbits’ behavior was first analyzed in the 6-pen trial setup for 4 weeks and then for additional 4 weeks in the 3-pen trial setup. We analyzed 30 min of video material to define the baseline behavioral repertoire at the first three days of the initial transfer to the 6-pen trial setup (overall 90 min of video material). Further, we analyzed animals’ behavior in the 6-pen/3-pen trials by analyzing a total of 240 min of video material.

### 2.5. Statistical analysis

Parametric data are presented as mean ± standard error of the mean (SEM) or as median and interquartile range (IQR) for non-parametric data. Statistics were calculated using the software GraphPad Prism (Version 5; GraphPad, San Diego, CA, USA). The Kolmogorov–Smirnov test was used to test for Gaussian distribution. The Kruskal–Wallis test followed by Dunn’s multiple comparison was used to calculate *p*-values of non-parametric data. One-way repeated measure ANOVA was used to identify significant differences between the two trials (6-pen vs. 3-pen trial) tested. Two-way repeated measure ANOVA was used to identify statistically significant behavioral differences among different time points and trials tested. *p* < 0.05 was considered statistically significant if not stated otherwise.

## 3. Results

We tested our mobile and modular floor-pen housing system (named: mobile-R-pen) under specific-pathogen-free housing conditions in two different experimental setups. First, we evaluated the health status in twelve female New Zealand White rabbits over a period of 12 months while they were housed in the mobile-R-pen. Second, we evaluated the species-specific circadian behavior of an additional six female New Zealand White rabbits for a total of 8 weeks. Furthermore, we evaluated the animals’ behavior after size restriction in two trials with different sizes of the enclosure.

Health study: During the health study, rabbits were single- or group-housed on loose wood chip bedding material. Wood chip bedding was used to reduce bone deformations or paw injuries (Figure 2A,B). Animals did not show any bone deformations or paw injuries at the end of experimentation. The SPF status of the colony was stable during the study, indicating adequate and appropriate decontamination measures of the floor-pen elements.

Behavioral study: The behavioral repertoire of six young adult female NZW rabbits was evaluated in two floor-pen setups differing in the size of the enclosure. We analyzed rabbits’ behavior during an 8-week period based on the ethogram depicted in Table 1. Interestingly, rabbits spent significantly more time hiding in the 6-pen trial compared to the 3-pen trial, and showed a trend towards higher rearing, chewing, and self-grooming activities in the 3-pen trial compared to the 6-pen trial, though no statistically significant differences were reached in those three behaviors (Table 2).

Furthermore, rabbits showed a broad range of species-specific behavioral patterns in the 6-pen and the 3-pen trials (Figure 3). While there was significant variation between the two trials in the amount of time that the rabbits engaged in some behaviors (Table 2), the total frequency of events for each behavior showed no significant differences between trials (Figure 3). In the 3-pen trial, we observed an increase in the number of grooming, burrowing, fighting, and hiding behavior in comparison to the 6-pen trial, while the number of exploratory events dropped (Figure 3).

Next, to investigate whether the mobile-R-pen allows the performance of circadian behavioral repertoires, we counted rabbit-specific behavior at four different time points between 7:30 a.m. and 12:30 p.m. during 4 weeks of housing in the 6-pen/3-pen trials (Figure 4A,B). Importantly, rabbits showed circadian behavioral patterns in both housing setups (Figure 4A,B). Furthermore, we could identify significant differences in behavioral patterns at different time points in both trials tested (Figure 4C).

## 4. Discussion

In this study, we implemented a novel, mobile housing system (mobile-R-pen) for rabbits that provides spatial flexibility. Our analyses revealed species-specific behavioral patterns of group-housed female NZW rabbits in a larger enclosure as well as in size-restricted compartments. Size restriction led to no general alteration of NZW rabbits. We observed that hiding was significantly decreased with size restriction. Moreover, fighting was increased in 6-pen versus 3-pen trials, indicating enhancing species-specific behavior that accompanies hierarchy establishment. In line with this notion is the finding that size restriction is a stressor that impacts social structure establishment in female NZW rabbits [17]. The time animals spent hiding was significantly increased in the 6-pen trial, while the number of hiding events was lower in the 3-pen trial. Interestingly, the time animals spent hiding decreased after size restriction of the already familiar environment. This might be an indication that social stress affects animals and prevents them from resting for longer periods of time. It has been reported that self-grooming increases after size restriction, while explorative behavior increases with enclosure size for NZW rabbits [15]. We demonstrate that the time animals spent exploring the enclosure drops in parallel to the number of exploratory events in the 3-pen trial compared to the 6-pen trial. Thus, space affects animal behavior and, in turn, the outcome of experiments. Our data are in line with recent reports indicating that the size and shape of the enclosure are critical variables that might induce stressful situations for group-housed rabbits [16]. Of note, the floor size of a single mobile-R-pen module exceeds the minimal space recommendations of the European directive by 2.3 times, allowing optimal adaptations for young and adult rabbits [1].

Moreover, we identified fighting events, but no injuries, which has been observed after re-grouping of rabbits in an unfamiliar pen by others [18]. Furthermore, we did not identify fighting events during our baseline behavior analyses in the 6-pen trial, which might indicate that first introducing animals to a larger, unfamiliar environment reduces aversive behavior prior to size restriction [17]. The time animals spent fighting was significantly different in the 6-pen trial compared to the 3-pen trial in that the number of fighting events increased after size restriction. This indicates that sufficient space promotes the establishment of social hierarchy. Moreover, space restrictions led to increased fighting, most likely due to special stress. Sufficient space to escape and adequate enrichment items to hide in or behind might therefore improve animals’ well-being and prevent injuries and intense fights, even after re-grouping [19].

Next, we identified a circadian behavioral pattern of group-housed rabbits, independent of the size of the enclosure. Our data mirror the natural behavior of *Oryctolagus cuniculus* even under standardized conditions in a vivarium. In line with recent observations [20], we identified a peak of activities in the early morning, which continuously dropped until midday. This should be further considered when planning and performing animal experiments and interpreting data. Interestingly, neither the 6-pen trial nor the 3-pen trial had an impact on species-specific and circadian behavioral pattern. Moreover, our setup prevents unnecessary interactions of animals and caretakers that might disturb group stability [4]. Even in situations that favor single-housing over group-housing in research experiments, so-called social non-contact enrichment should be implemented [21]. In this respect, the mobile-R-pen housing concept allows visual, auditory, and olfactory communication as well as opportunities to escape the direct contact of neighboring individuals due to transparent and opaque sub-dividing walls and adequate enrichment. The major improvement of the mobile-R-pen is the easier separation of individuals and re-grouping and co-housing of animals, depending on the weight of the individuals or size of the colony, without the need for additional handling of animals. This is an important refinement factor in research experimentation to minimize stressful situations.

This system is applicable in animal facilities where high hygiene standards are important. In contrast to commercially available floor-pen systems, the mobile-R-pen concept allows a maximum of flexibility in the interior design of a vivarium without the need for a stable connection to the floor or wall of the room. Mobile-R-pens can be easily moved or transferred (with or without rabbits) for experiments or for decontamination and cleaning. This is a major advantage of the housing system that contributes to the minimization of hygiene risk and disease spread. The solid ground of the pen allows the use of autoclavable bedding materials normally used in most mouse/rat animal facilities. This soiled bedding material can be easily removed when necessary and allows the monitoring of excretion from outside of the pen. The health status of the rabbits was not negatively affected by single or group housing and contact with soiled bedding materials, which is in line with previous reports [22]. It is a major advantage of the mobile-R-pen that a sufficient amount of bedding material can be provided to prevent bone deformation or severe skin lesions. In comparison to traditional cage housing, where bedding material is commonly not used, the use of loose material allows species-specific behavior, like burrowing, that contributes to the reduction of stereotypies and improvement of the animals’ well-being [15]. Future studies are clearly needed to investigate the beneficial impact of loose bedding material in our housing system.

## 5. Conclusions

In conclusion, easy handling and disassembly of pens ensure sufficient decontamination as an important measure for SPF housing conditions of rabbit colonies. Our improved mobile-R-pen housing concept supports species-specific, circadian behavioral repertoire, which is of great benefit for the well-being of single- and group-housed laboratory rabbits and provides an easy refinement concept according to animal welfare recommendations for rabbits used in research.

## Figures and Tables

**Figure 1 animals-11-00977-f001:**
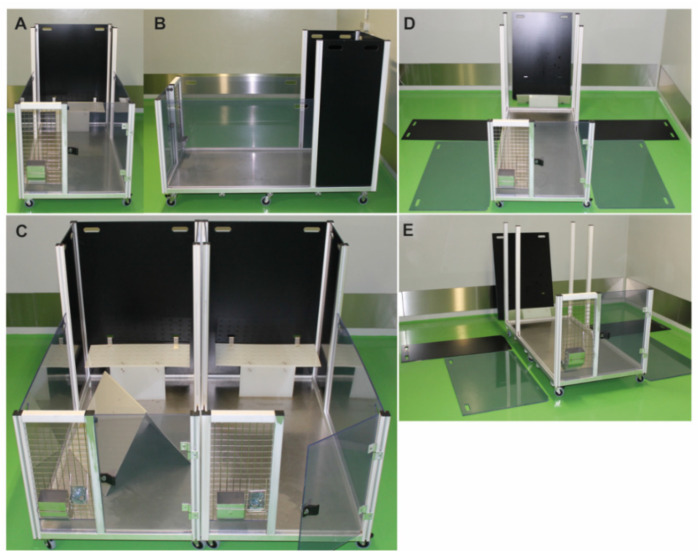
Mobile-R-pen housing equipment front view (**A**) and lateral overview (**B**). Water bottles, food containers, or additional enrichment can be added to the gridded front panel of each module. Transparent and black side walls separate individual pen elements and can be removed to connect two or more mobile-R-pen modules to a larger area (**C**). Each pen consists of five surrounding walls that can be completely removed from the scaffold, which is mounted on six wheels (**D**,**E**).

**Figure 2 animals-11-00977-f002:**
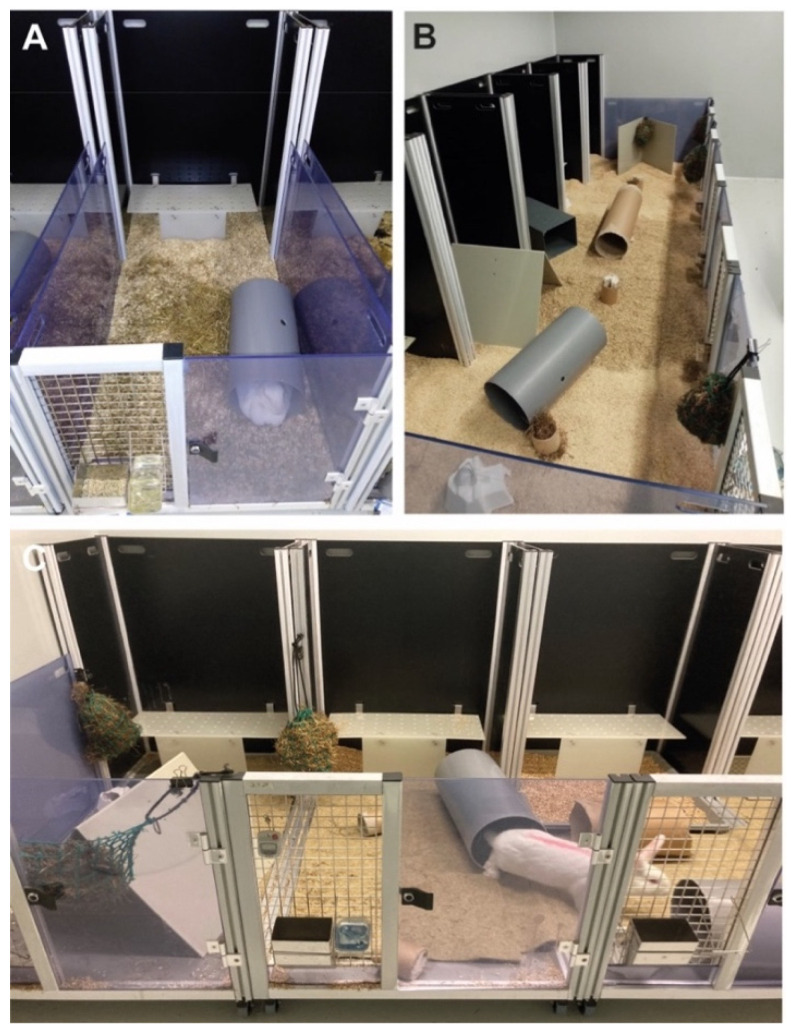
Set-up for a single-housed rabbit (**A**) or group-housed animals (**B**,**C**) in a mobile-R-pen. Modules can be assembled to provide flexibility in defining housing conditions. Each module consists of an elevated platform that is mounted on the backplane of the scaffold. Additional enrichment items (e.g., tents, tubes, hay nets) can be added to individual modules as depicted in (**A**–**C**).

**Figure 3 animals-11-00977-f003:**
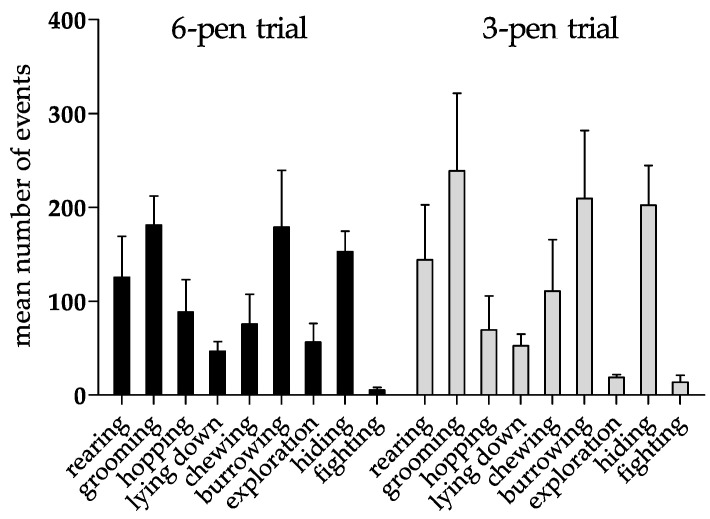
Behavioral pattern of six rabbits first investigated in the 6-pen trial for four weeks and afterwards in the 3-pen trial for an additional four weeks. Data are displayed as arithmetic means + SEM. Mean number of events counted over a period of four weeks in the 6-pen/3-pen trials (*n* = 6 animals per trial). Parametric data were analyzed by one-way repeated measure ANOVA. The test revealed no significant differences between different trials.

**Figure 4 animals-11-00977-f004:**
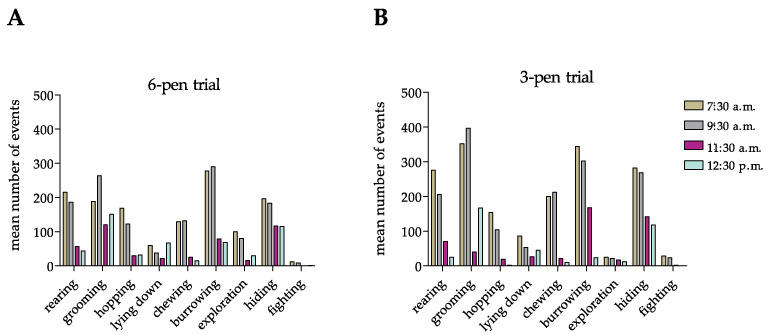
(**A**,**B**) Rabbits’ circadian ethogram during the 6-pen trial and 3-pen trial. Data are displayed as arithmetic means (**A**,**B**) or means + SEM (**C**). Mean number of events was counted over a period of four weeks in the 6-pen/3-pen trials (*n* = 6 animals per trial). Parametric data were analyzed by two-way repeated measure ANOVA to identify significant differences between time points and trials (**C**). Significant differences between time points compared to 7:30 a.m. are indicated by an asterisk (*p*-value < 0.05).

**Table 1 animals-11-00977-t001:** Rabbit ethogram.

Rabbit Behavior Definition	
Rearing	Animal is up on both hind legs and torso is perpendicular to the floor
Grooming	Animal is licking or scratching its coat or face
Hopping	Animal is moving forward by pushing hindlimbs followed by forelimbs
Lying down	Animal is in a horizontal position on an elevated platform
Chewing	Animal is actively biting on non-food material
Burrowing	Animal is using forepaws to dig into the bedding material and to scratch on the floor of the pen
Eating	Animal is biting and swallowing food
Drinking	Animal is consuming water
Exploration	Animal is actively moving around, jumping, and sniffing on enrichment material
Hiding	Animal is sitting/lying under or behind tent/tube/platform or opaque wall element
Fighting	Animal is biting, chasing, mounting, or snapping at other rabbits
Other	Animal performs other behavioral patterns than those stated above. The value is calculated as the difference of all measured behaviors from the overall time of observation

**Table 2 animals-11-00977-t002:** Median time spent (in seconds) performing different behavioral patterns.

Rabbit Behavior	Baseline (a)		6-Pen Trial (b)		3-Pen Trial (c)		*p*-Value
	Median	IQR	Median	IQR	Median	IQR	K–W Test
Rearing	7.5	29.25	7.5	13.5	15.0	18.0	n.s.
Grooming	36.0	47.25	15.0	46.5	36.0	69.0	n.s.
Hopping	13.5	11.25	12.0	21.0	6.0	7.5	n.s.
Lying down	30.0	0.0	84.5	481.5	279.0	204.0	n.s.
Chewing	12.0	6.0	16.5	3.0	49.5	94.5	n.s.
Burrowing	7.5	38.5	42.0	42.0	19.5	48.0	n.s.
Eating	355.5	756.8	567.0	804.0	594.0	609.0	n.s.
Drinking	426.0	648.0	246.0	294.8	91.5	354.7	n.s.
Exploration	45.0	82.5	96.0	127.5	39.0	58.5	n.s.
Hiding	859.5	1523.2 ^(a,c)^	510.0	801.0 ^(b,c)^	261.0	474.7	0.0432
Fighting	0.0	0.0 ^(a,c)^	28.5	71.25 ^(b,c)^	16.5	34.5	0.0001
Other	7.5		175		393		

Baseline (a): First three days after the initial transfer to the 6-pen trial. 6-pen trial (b): Four weeks of group housing of rabbits (*n* = 6) in 6 connected pens. 3-pen trial (c): Four weeks of group housing of rabbits (*n* = 6) in 3 connected pens previously housed in the 6-pen trial. Data are displayed in seconds (median and interquartile range (IQR)). Other: Calculated as the difference of all behavioral patterns shown from the overall time of investigation per trial (30 min). Kruskal–Wallis (K–W) test followed by Dunn’s multiple comparison (non-parametric data) was used to determine the *p*-values between baseline and 6-pen/3-pen trials. Superscript letters (a, b, c) indicate the groups that were compared and showed a significant difference (baseline (a), 6-pen trial (b), or 3-pen trial (c)) (*p*-value < 0.05). n.s. = no significant difference between the two trails and baseline.

## Data Availability

All relevant data is given in the paper. Additional information can be requested from the authors upon reasonable request.

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
