# Peer review of "Evaluation of a Configurable, Mobile and Modular Floor-Pen System for Group-Housing of Laboratory Rabbits"

_animals, 2021, doi:10.3390/ani11040977_

Round 1

Reviewer 1 Report

This is a resubmission of a paper I previously reviewed on a new housing systems for rabbits. The authors have made substantial revisions in line with the feedback given to them. However, aspects of experimental design are still unclear and I think the results are overstated. The message being sent out is also unclear and I think the authors should explicitly state their study aim and restrict their discussions to this aim. The focus of the study appears to be on the effects of space on selected health and behavioral outcomes, not on comparison between traditional system or cleaning ability. The manuscript needs revising considering the significance levels reported in Table 2.

The methods have been much improved and are clearer- although I still think a specific statement of the aim which matches the experimental design is needed. The data suggests that you are comparing space allocation (3 pent trial vs 6 pen trial) although it is unclear whether group size (l168) has been controlled and hence this may be a confounder. 

Line 178- I'm still not clear on the distribution of the 6 rabbits for behavioural sampling across the 2 test conditions. This number seems a little low as a sample size given the inherent variability in behavioural outcomes. How did you derive it as a sample size? This may have resulted in the non-significances seen. 

Lines 267-274 and paragraph above- it is stated that in the '3-pen-trial, we observed an increase in the number of grooming, burrowing, fighting and
 hiding behavior in comparison to the 6-pen-trial' but Table 2 only notes significant differences for fighting and hiding. The para above similarly talks about 'higher rearing, chewing and self-grooming activities but there is no significance in the table. 

L334- I agree that loose bedding may reduce stereotypy formation but you haven't specifically demonstrated this so would include that this needs examination in your system as compared to traditional systems. 

L351- I'm not clear that your results show general changes in specific behavioural patterns due to the lack of significance between conditions . Hiding is the only one that was significant.  

The conclusion is unusual in starting by discussing decontamination which is not the focus of your study. The conclusion should relate to what you have investigated. 

Author Response

This is a resubmission of a paper I previously reviewed on a new housing systems for rabbits. The authors have made substantial revisions in line with the feedback given to them. However, aspects of experimental design are still unclear and I think the results are overstated. The message being sent out is also unclear and I think the authors should explicitly state their study aim and restrict their discussions to this aim. The focus of the study appears to be on the effects of space on selected health and behavioral outcomes, not on comparison between traditional system or cleaning ability. The manuscript needs revising considering the significance levels reported in Table 2.

Authors response: Thank you for raining these points. We further specified the aim of our study and restricted our discussion to this aim. For revising the experimental design, we followed the suggestions of this referee. Further, we revised the data presentation in table 2. We now include the p-values for of the Kruskal-Wallis test (indication for differences in rank sums in-between the three groups). Moreover, using superscript letter, we show which groups have been compared and whether there is a significant differences between them. We included statistics and description in the figure legend. We hope table 2 is clearer now.

The methods have been much improved and are clearer- although I still think a specific statement of the aim which matches the experimental design is needed. The data suggests that you are comparing space allocation (3 pent trial vs 6 pen trial) although it is unclear whether group size (l168) has been controlled and hence this may be a confounder.

Authors response: Thank you for pointing this out. We specified the aim of our study at the end of the introduction. For our space allocation experiment, we either housed two groups of two animals and one group of 3 animals in 3 pens (2x2+3) or one group of five animals in 6 pens. Thus, our sample size is 7 for 3-pen trials and 5 for 6-pen-trials. Therefore, the sample size is fairly similar. We corrected the number of animals in the methods section specified this information in the text.

Line 178- I'm still not clear on the distribution of the 6 rabbits for behavioural sampling across the 2 test conditions. This number seems a little low as a sample size given the inherent variability in behavioural outcomes. How did you derive it as a sample size? This may have resulted in the non-significances seen.

Author response: Thank you for raising this point. For our behavior tests, we focused on a study by DiVincenti et al., J Am Assoc Lab Anim Sci 2016. In this study, DiVincenti et al. stated, that even wild rabbits tend to socialize in groups of 3 or fewer. For animal welfare reasons, we did not include additional animals. Future studies are clearly needed to investigate the impact of size restrictions on animal behavior and to spot differences between conventional and our modular floor pen housing system.

Lines 267-274 and paragraph above- it is stated that in the '3-pen-trial, we observed an increase in the number of grooming, burrowing, fighting and hiding behavior in comparison to the 6-pen-trial' but Table 2 only notes significant differences for fighting and hiding. The para above similarly talks about 'higher rearing, chewing and self-grooming activities but there is no significance in the table.

Author response: Thank you for pointing this out. We rephrased this part.

L334- I agree that loose bedding may reduce stereotypy formation but you haven't specifically demonstrated this so would include that this needs examination in your system as compared to traditional systems. 

Author response: We agree. We toned down our statement. L. 390-391 of the revised manuscript.

L351- I'm not clear that your results show general changes in specific behavioural patterns due to the lack of significance between conditions. Hiding is the only one that was significant.  

Author response: Thank you for your comment. We observed that fighting and hiding were significantly affected. This referee is right in saying that we do not observe general changes. We rephrased the discussion to tone down our conclusions.

The conclusion is unusual in starting by discussing decontamination which is not the focus of your study. The conclusion should relate to what you have investigated.

Author response: Thank you for your comment. We restructured the discussion part.

Reviewer 2 Report

This manuscript has been considerably improved from the first one. The structure is much clearer, and there are no misplaced methods or results. I am happy to recommend it for publication, pending a few further minor edits that will hopefully clarify a few confusing elements.

L46 - suggest changing 'stressful' to 'stress-related'.

The first para of the intro is very long. Suggest cutting into two or three smaller ones to improve flow.

L118 - when the authors say 'lucent', do they mean 'translucent'? 'Lucent' means 'giving off light', which 'translucent' means 'see-through'.

Table 2 - p-value column. Which comparison is this referring to? Ideally there would be three columns for the p-values for all three comparisons

L269-270 - this sentence is unclear, and it's also two sentence fragments. It should be made clearer that table 2 is referring to time spent engaging in the behaviours, and fig 3 refers to total frequency of behavioural events. Suggest changing to something like: 'While there was significant variation between the two trials in the amount of time that the rabbits engaged in some behaviours (Table 2), the total frequency of events for each behaviour showed no significant differences between trials (fig 3).'

Author Response

This manuscript has been considerably improved from the first one. The structure is much clearer, and there are no misplaced methods or results. I am happy to recommend it for publication, pending a few further minor edits that will hopefully clarify a few confusing elements. 

L46 - suggest changing 'stressful' to 'stress-related'.

The first para of the intro is very long. Suggest cutting into two or three smaller ones to improve flow.

L118 - when the authors say 'lucent', do they mean 'translucent'? 'Lucent' means 'giving off light', which 'translucent' means 'see-through'.

Authors response: We agree with this referee. We changed wording in L.46 and L122. In addition, we cut the first paragraph into three parts.

Table 2 - p-value column. Which comparison is this referring to? Ideally there would be three columns for the p-values for all three comparisons

Authors response: We revised the data presented in Table 2. Significant levels have been determent by non-parametric Kruskal-Wallis test followed by Dunn’s multiple comparison. We show now the p-values for the Kruskal-Wallis test. Moreover, we show superscript letters indicating significant differences between the groups tested (differences in-between baseline, 6-pen-trial and 3-pen-trial). We chose this data presentation for the sake of clarity. We hope that this referee will agree.

L269-270 - this sentence is unclear, and it's also two sentence fragments. It should be made clearer that table 2 is referring to time spent engaging in the behaviours, and fig 3 refers to total frequency of behavioural events. Suggest changing to something like: 'While there was significant variation between the two trials in the amount of time that the rabbits engaged in some behaviours (Table 2), the total frequency of events for each behaviour showed no significant differences between trials (fig 3).'

Authors response: We agree. We changed the sentence according to the suggestion. L. 274 – L.277 of the revised manuscript.

Reviewer 3 Report

 The authors did a good and extensive editing job and correcting the MS and answer all reviewer's comments and the MS now in an acceptable form that may be accepted by the EiC.

Author Response

The authors did a good and extensive editing job and correcting the MS and answer all reviewer's comments and the MS now in an acceptable form that may be accepted by the EiC.

Authors response: We would like to thank you for the very positive feedback.

Round 2

Reviewer 1 Report

Thankyou for responding to these points. 

This manuscript is a resubmission of an earlier submission. The following is a list of the peer review reports and author responses from that submission.

Round 1

Reviewer 1 Report

This report presents an investigation of  a new housing system for laboratory rabbits with some evaluation of the behavioural repertoire of the animals over a period of housing, as well as a determination of pathology related to fractures. I believe that this could be a useful paper although it read a little between a sales pitch for a new caging system and a scientific article. My concern with the article is around the clarity of the aims/project itself which made it hard for me to ascertain whether the conclusions were actually valid.  For instance there was discussion about the issues of group versus single housing in rabbits and effects of different space allowances and there was no clear statement of the study aims. I assumed the idea was to compare singly housed versus group housed animals but later on the discussion seemed to focus more on space allowance. If this was the case then I'm not sure how it could be concluded that the system will benefit wellbeing since it was not compared to any other traditional housing system. Increased clarity in the aims/direction and methodological details is required before I could recommend this proceeding to publication. Some specific queries are below:

Introduction: Some discussion of the issues with rabbit housing in research would be beneficial, for example challenges with fighting which is why single housing may be used, differences between males and females. Perhaps some background on how rabbits are currently housed (albeit general) to set the scene for the new caging system. 

I think a clear aim for the study is needed at the end of the introduction.

Line 90- do the EU guidelines really suggest a max floor space- wouldn't this be a minimum?

L96- typo "hidding'

The experimental design was really unclear to me. There were 18 animals with 12 used for necropsy and only 6 for behavioural observations? or were all 18 used for behaviour. How was this allocated between groups and what were the groups. What was your experimental unit? You mention group sizes ranging from 2-3 but this (depending on your aim) may confound your study since this impacts allocation per rabbit if this is not consistent across the groups, plus social dynamics will be different. Perhaps a flowchart of the design would be helpful. 

Discussion on cage set up, pictures and rabbit ethogram needs to go into methods from results. 

Did you use a focal or scan sampling method?

Statistics: experimental unit especially for behavioural observations which may be at pen level. Which test of normality did you use. You also need to state here or later in results which test ended up being used on the data presented. 

L156- 'Wood chip bedding material was preferentially used to reduce bone deformations or paw injuries'. I'm not sure how you can back this up with the data collected since there is not control with wood chip. 

Results: this was hard to assess as I wasn't clear on the aims and what the comparator was. For example you talk about burrowing l158 'Furthermore, the loose bedding material on enables extensive burrowing behavior' but in comparison to what?

Table 2 needs the units adding and sample sizes as its not clear from above. 

If the behavioural data were analysed non parametrically it may be better to visualise the data as medians with an indication of dispersion. 

Fig 3: typo 'trail'

Reviewer 2 Report

This manuscript describes a new way to increase environmental enrichment afforded to laboratory rabbits via a species-appropriate housing system that also meets hygiene standards in specific-pathogen-free facilities. It represents an exciting new development in lab animal housing, and it has potential to be very important. As written, though, I cannot recommend it for publication.

My major concern is that the methods and results are jumbled and unclear. Some of the methods are presented in the results, and the analyses described in the methods are not sufficient to understand what exactly was analysed. Furthermore, the intro is very short, which is fine, but it does not fully justify all analyses done. For instance, why do we care about changes in behaviour due to circadian rhythm? This is not explained anywhere so it comes out of nowhere in the results section.

The authors appear to assume too much knowledge on the part of the reader. Acronyms like SPF, IVC, and FELASA-14 are not defined anywhere. NZW is defined in the methods section, but it is first mentioned in the intro. Definitions should be provided where they are first mentioned in text.

Specific recommendations

Simple summary - this is very confusing as written. The abstract is written much more clearly. 'novel, self-invented concept for single...' should be replaced with ' novel housing system for single...' It is also not clear exactly what 'specified-pathogen-free' means. Suggest using wording more appropriate for a lay audience, as that is what the simple summary is supposed to be. Finally, at the end of the simple summary, suggest changing '..those housing concept' to 'this housing system'

Abstract - suggest clarifying that six additional rabbits were tested for behaviour, in addition to the original 12 in the health study, not a subsample of the 12 rabbits in the health study.

Keywords - in general, the keywords should be words that are not already in the title, as words in the title will automatically be indexed for searches, as I understand it. I therefore suggest removing 'floor-pen', 'rabbits', 'group-housing'. I also recommend changing 'refinement' to something more specific, as refinement is very broad and not limited to animal welfare issues. '3Rs' or '3R strategy' would be better. I'd also add 'animal welfare' as a keyword.

Intro - please define NZW and SPF. Also, the last sentence in the intro should be moved to the Discussion. Please better justify the need to study circadian rhythm impacts on the rabbits.

Methods - L71-80 'Room temperature...working group' should be moved into a different section called 'Lab environment' or similar.

L80-L86 should be moved to procedure section.

All experiments were in accordance with authorities, but is there an ethics approval number for this study? If not, why not?

L84 - 'government of upper bavaria' should be 'Government of Upper Bavaria'.

L95 - suggest changing 'self-made tubes and tents' to 'tubes and tents made by the research team'.

Figures and tables should be listed just below where they are referred to in text. Figs 1 and 2 are referred to in the Housing para of the methods, but aren't shown until the results section on the following page. Suggest moving them higher up, just under L101.

Suggest changing 'concept of the study' to 'procedure', to follow standard scientific reporting protocol. 

L115 - so the study on the 6 rabbits was a repeated measures study, whereby all six were allocated to both conditions? Clarify. How long did they remain in each pen size?

Statistics subsection - more info is needed on what was analysed. The statistical tests were described but not what variables they were used to test. Also, which data are non-parametric? Are all of them non-para? If so, say so. If not, explain which ones are para and which ones non-para.

I suggest restructuring the methods section slightly with the following subheaders:

  1. Animals
  2. Laboratory Environment
  3. Housing
    1. Standard
    2. Small mobile-R-pen
    3. Large mobile-R-pen
  4. Procedure
    1. Health study
    2. Behavioural study
  5. Statistical analyses
    1. Health study
    2. Behavioural study

Results

The results should not be describing the housing system or the procedure. They should just present the results of the health and behavioural tests with the rabbits.

In line with my recommended structure changes in Methods, I suggest doing similar with the Results, with two subsections, one for the Health Study and one for the Behavioural Study.

L134-136 - 'the modular character of ...' this belongs in the Discussion

L136-138, L145-155, and L158-162 belong in the methods section, in housing description.

L169-175 should be in the procedure section of the methods.

Table 1 - 'non-lucent' would be better as 'opaque'. Check through text for other places where 'non-lucent' is used and change to opaque.

Table 1 - 'forwarded' should be 'forward'. In many rows, 'animals' is used instead of 'animal'.

Suggest modifying Table 2 caption to 'Average time spent (in sec) performing different...' to make it clear to the reader that the results presented are in seconds. It's in the note below the table but it's buried in there and it should be immediately obvious.

Table 2 - since there is an 'other' category, I am confused why the average times for the different behaviours is so low. In 30 minutes, there would be 1,800 sec. But the average time spent engaged in all of the behaviours combined for the baseline was 379.1sec. Even taking the SD into account, the total is nowhere near 1800. I don't expect it to add up to 1800sec precisely because they are averages, but less than 25% of that total is accounted for, which doesn't make sense. What were the rabbits doing that was not represented in the table?

L209 - restate that each data collection period was 30 minutes.

Fig 3 - why did the authors not run a within-rabbit, between-conditions statistical analysis (e.g. a mixed model ANOVA) to compare the rabbits' behaviours in the two conditions? That would be an instructive addition to the frequency data presented in the Figure.

Discussion

L237-239 - check grammar of this sentence. It reads like a run-on. A comma or two, or separating the sentence into two, should help.

The first para of the Discussion is too long. Suggest cutting into 2 or 3 to improve flow.

L255 - suggest removing the word 'supportively'.

L290-294 - why is circadian rhythm relevant here? It has not be justified in the intro.

General - there are minor spelling and grammar errors throughout. For instance, 'stereotypes' should be 'stereotypies', and words like 'hierarchy' and 'withdrawal' are misspelled in the intro. 'might' and 'favor' are misspelled in the Discussion.

Also, there are many times when "animals behaviour" is written, but it should be "animals' behaviour". I suggest putting the document through the MS Word spellcheck and grammar checker. It has improved considerably in recent years and, in my experience, with rare exception (e.g. stereotypies may look like a spelling error in Word), if it indicates an error, it's because there is one. This is especially true for grammar errors.

Reviewer 3 Report

 See the attached version, a few comments can be made  and  the following should be considered 

1- Several spelling mistakes were noticed in the text and indicated in the attached copy. 

2- I would rather prefer Tukey instead of Duncan's for statistical analyses.

3- I would also like to adjust the space per rabbit to per kg housed to avoid differences in body weight.
